# The Males Absent on the First (MOF) Mediated Acetylation Alters the Protein Stability and Transcriptional Activity of YY1 in HCT116 Cells

**DOI:** 10.3390/ijms24108719

**Published:** 2023-05-13

**Authors:** Tingting Wu, Bingxin Zhao, Chengyu Cai, Yuyang Chen, Yujuan Miao, Jinmeng Chu, Yi Sui, Fuqiang Li, Wenqi Chen, Yong Cai, Fei Wang, Jingji Jin

**Affiliations:** School of Life Sciences, Jilin University, Changchun 130012, China; ttwu18@mails.jlu.edu.cn (T.W.); bxzhao20@mails.jlu.edu.cn (B.Z.); caicy1322@mails.jlu.edu.cn (C.C.); yyc20@mails.jlu.edu.cn (Y.C.); yjmiao19@mails.jlu.edu.cn (Y.M.); chujm@mails.jlu.edu.cn (J.C.); suiyi0910325@163.com (Y.S.); lifq20@mails.jlu.edu.cn (F.L.); chenwq1313@mails.jlu.edu.cn (W.C.); caiyong62@jlu.edu.cn (Y.C.)

**Keywords:** MOF, histone acetyltransferase, cell proliferation, transcription factor, Yin Yang 1

## Abstract

Yin Yang 1 (YY1) is a well-known transcription factor that controls the expression of many genes and plays an important role in the occurrence and development of various cancers. We previously found that the human males absent on the first (MOF)-containing histone acetyltransferase (HAT) complex may be involved in regulating YY1 transcriptional activity; however, the precise interaction between MOF-HAT and YY1, as well as whether the acetylation activity of MOF impacts the function of YY1, has not been reported. Here, we present evidence that the MOF-containing male-specific lethal (MSL) HAT complex regulates YY1 stability and transcriptional activity in an acetylation-dependent manner. First, the MOF/MSL HAT complex was bound to and acetylated YY1, and this acetylation further promoted the ubiquitin–proteasome degradation pathway of YY1. The MOF-mediated degradation of YY1 was mainly related to the 146–270 amino acid residues of YY1. Further research clarified that acetylation-mediated ubiquitin degradation of YY1 mainly occurred through lysine 183. A mutation at the YY1K183 site was sufficient to alter the expression level of p53-mediated downstream target genes, such as *CDKN1A* (encoding p21), and it also suppressed the transactivation of YY1 on *CDC6*. Furthermore, a YY1K183R mutant and MOF remarkably antagonized the clone-forming ability of HCT116 and SW480 cells facilitated by YY1, suggesting that the acetylation–ubiquitin mode of YY1 plays an important role in tumor cell proliferation. These data may provide new strategies for the development of therapeutic drugs for tumors with high expression of YY1.

## 1. Introduction

Yin Yang 1 (YY1), a well-known transcription factor with multiple zinc finger domains, was first discovered as a DNA-binding protein. It belongs to the GLI-Kruppel family, and it is highly conserved from Xenopus to humans [1,2]. YY1 contains both transcriptional activation and transcriptional repression domains, which have versatile functions in gene transcription regulation, according to the specific cofactors recruited [3]. According to published literature, YY1 includes two repression domains—one embedded within residues 170–200 and the other overlapping with the C-terminal zinc finger DNA-binding domain—while the activation domain is located in the N terminal [4]. Therefore, YY1 is widely involved in various biological functions in cells, including cell growth, development, differentiation, and apoptosis [5]. The roles of YY1 are controlled by protein–protein interactions, and many transcription factors or cofactors have been revealed to associate with YY1. The interactions between YY1 and p300, as well as between YY1 and histone deacetylases (HDACs), have been extensively reported. As a case in point, p300 HAT physically interacts with YY1, reducing YY1 transcriptional repression by adenovirus E1A [6]. In certain circumstances, YY1 binds to HDACs. For example, HDAC1/2 associates with residues 170–200 of the suppression domain of YY1 to exert transcriptional inhibitory effects [4]. Taken together, by selectively binding to HATs or HDACs, YY1 becomes an activator or suppressor. However, it has not yet been clearly determined how YY1 actively chooses its binding partner.

MOF, a member of the MYST HAT family, participates in various important biological processes in cells [7]. MOF serves as a catalytic subunit to form a male-specific lethal (MSL) complex that is highly conserved in structure and composition from *Drosophila* to humans [8]. In *Drosophila*, the MSL complex is required for dosage compensation [9]. Except for the catalytic subunit MOF, MSL1–3 subunits exist in both *Drosophila* and human cells. As an integral scaffold protein, the N-terminal of MSL1 forms a bridge link with MSL2, while the C-terminal of MSL1 connects with MOF and MSL3 [10]; therefore, it can be inferred that MSL1 plays a crucial role in maintaining the holoenzyme activity of the overall complex. Importantly, the main functions of the MSL complex are closely related to the enzymatic activity of the catalyzing histone H4K16ac [8].

Human MOF (also known as KAT8 or MYST1) is a 458 amino acid (aa) protein that contains a conserved MYST catalytic domain, a chromodomain, and a C2HC-type zinc finger [11], and it forms two distinct complexes: MSL and NSL. Based on recent reports, the two complexes are involved in the epithelial–mesenchymal transition (EMT) in different ways in human cells. Silencing the MSL complex promotes the EMT process; however, silencing the NSL complex inhibits the EMT process [12]. Multi-omics studies further clarified that the MSL complex is more inclined to regulate biological processes related to cell morphology, skeleton, migration, proliferation, and division [12]. The roles of human MOF-mediated H4K16 acetylation in DNA damage repair, cell development, and tumorigenesis have been widely reported [13,14,15,16]. In addition to their involvement in transcriptional regulation through H4K16ac, MOF complexes can also acetylate non-histone proteins, such as transcription factors p53, nuclear factor-erythroid factor 2 (Nrf2), and TTF–I interacting protein 5 (TIP5) [17,18,19]. Interestingly, transcription factor YY1 was identified as a transcriptional target of MOF-containing complexes. Chromatin immunoprecipitation sequence (ChIP-Seq) analysis showed that the peaks of MOF co-localized with H4K16ac, H3K4me2, and H3K4me3 at the transcriptional start site of YY1. Furthermore, silencing the NSL complex suppressed the clonogenic ability of HepG2 hepatocellular carcinoma cells [20]. However, the precise mechanism by which MOF-containing complexes regulate YY1 is still unclear.

It has been reported that YY1 undergoes post-translational modifications (PTMs), including phosphorylation, ubiquitination, acetylation, O-GlcNAcylation, S-nitrosation, sumoylation, and poly(ADP-ribosyl)ation [21,22,23,24,25,26,27]. As a key means to regulate protein function, PTMs play an essential role by adding covalent groups to specific amino acid residues. Proteins are concurrently modified by numerous PTMs at multiple sites, often with one modification serving as the protein code, providing recognition markers for subsequent proteins to execute the next biological process. Therefore, different modifications often synergistically participate in regulating complex biological functions in cells [28]. For example, protein ubiquitination and acetylation share a fundamental link that is based on lysine residues and often involves crosstalk between two modifications to synergistically regulate key biological processes [29]. Previous studies have reported that the crosstalk between acetylation and ubiquitination plays an important role in regulating protein stability. The two intricate mechanisms controlling protein stability are: (1) protein ubiquitination-related biological processes that may be regulated by protein lysine acetylation; (2) the molecular basis of acetylation-mediated control of protein stability [30]. Herein, we first report that MOF was bound to and acetylated YY1, and this modification regulated ubiquitin-proteasome degradation of YY1 in human colon carcinoma cells. Using a gene mutagenesis approach and a series of biochemical and biological experiments, we identified the YY1K183/K258 sites as the main ubiquitin sites, and we showed that mutations at these sites affected the transcriptional regulation of YY1. Our findings offer a theoretical foundation for clarifying how YY1 functions in colon cancer cells.

## 2. Results

### 2.1. Interaction between YY1 and MOF Was Confirmed by Immunoprecipitation (IP) and GST-Pulldown Assays

In our previous study, we noticed that YY1 protein expression levels were regulated by NSL HAT [20]. In addition, MOF-containing complexes have been reported to be broad transcription regulators of constitutively expressed genes in both flies and mammals [31,32]. Therefore, we speculated that MOF and YY1 may interact and jointly regulate gene transcription. Thus, using a Flag IP approach, we first examined whether MOF and YY1 precipitated with each other. As shown in Figure 1A,B, overexpressed YY1 and MOF in HCT116 colon cancer cells immunoprecipitated endogenous MOF and YY1, respectively. To further confirm this result, co-transfection and Flag IP experiments were carried out. As expected, MOF appeared in the anti-Flag agarose eluates from Flag-YY1, suggesting the binding of YY1 with MOF (Figure 1C). The colocalization of endogenous YY1 and MOF was also confirmed by cell immunofluorescence (IF) staining (Figure 1D). Next, in order to verify the minimum binding region of YY1 with MOF, the deletion mutants of YY1 (1–146, 1–213, 1–270, 146–270, 146–414, 207–414, and 270–414) were subcloned into a pET41a vector, and His-GST tagged YY1 truncated proteins were expressed and purified (Figure 1E). A GST-pulldown assay was then performed by mixing whole cell lysate, prepared from Flag-MOF overexpressing 293T cells, and the GST-tagged YY1 deletion mutant proteins. The strongest binding was clearly between the YY1/146–270 region and MOF (Figure 1F). To further confirm this result, we analyzed the binding of YY1, without the 146–270 region, to MOF. As shown in Figure 1G, compared with wild-type YY1 (YY1wt), the binding of YY1/Δ146–270 to Myc-MOF was significantly reduced in HCT116 cells, demonstrating the importance of the 146–270 region of YY1 in the interaction between YY1 and MOF. To further confirm the interaction between MOF and the YY1/146–270 region, we designed a competitive binding experiment. As shown in Figure 1H, HCT116 cells were transiently transfected with Flag-YY1wt and Myc-MOF with or without YY1/146–270. Then, 48 h later, whole cell lysate was prepared, and the levels of MOF protein bound to YY1 were analyzed after a Flag IP. MOF was clearly bound to YY1; however, the amount of this binding combination decreased due to the addition of the YY1/146–270 region, resulting in a decrease in MOF binding to the full length YY1. Then, we analyzed the binding region of MOF with YY1. In HCT116 cells, Flag-YY1wt was co-transfected with Myc-tagged full length MOF, the N-terminal (1–157 aa), or the C-terminal (158–459 aa), and bound proteins were measured after a Flag IP. As a result, we concluded that YY1 interacted with the C-terminal of MOF (Appendix A).

### 2.2. Acetylation of YY1 by the MOF/MSL Complex Was Closely Associated with YY1 Ubiquitin-Proteasome Degradation in HCT116 Cells

As mentioned, we have shown that YY1 and MOF interact with each other. Given that the MOF/MSL complex is a HAT, we want to know whether MOF can acetylate YY1 and, if so, how the acetylation of YY1 impacts its protein stability. To address these questions, we first analyzed the expression of endogenous YY1, following the overexpression of wild-type MOF (MOFwt), or a MOFG327E mutant with the loss of enzyme activity in HCT116 cells. As shown in Figure 2A, two bands of YY1 were observed in cells transfected with MOFwt (upper panel, IB:YY1), but this phenomenon was not observed in overexpressing MOFG327E cells (lower panel, IB:YY1), suggesting that YY1 may be modified by MOF. A remarkable increase in Smurf2 protein levels was also observed in a transient transfection of MOFwt in a dose-dependent manner. Next, to determine whether YY1 can be modified by MOF, HCT116 cells were co-transfected with Flag-YY1 and Myc-MOF in the presence of HA-ubiquitin and the protease inhibitor MG132, and then, we tested whether YY1 was acetylated by MOF using a pan-acetylation (Pan-ac) antibody after a Flag IP (Figure 2B). As expected, a dose-dependent increase in YY1 acetylation was observed (Flag IP, IB:Pan-ac). Interestingly, the ubiquitin-mediated degradation level of YY1 also increased as the amount of MOF increased (Flag IP, IB:HA), suggesting that the acetylation of YY1 may negatively regulate its protein stability. To verify this result, we conducted repeated experiments in the presence or absence of the MOF HAT inhibitor MG149 [33]. As a result, the degradation of YY1 caused by MOF was inhibited by adding MG149, clarifying that the ubiquitin-mediated degradation of YY1 was, indeed, related to its acetylation by MOF (Figure 2C). To further confirm the involvement of MOF-mediated acetylation in the regulation of YY1 stability, we performed an experiment using the inactivated MOFG327E mutant. As shown in Figure 2D, compared to MOFwt transfected cells, YY1 acetylation was dramatically inhibited by the transfection of MOFG327E (Flag IP, IB:pan-ac). Decreased YY1 acetylation also led to a reduction in YY1 degradation (IB:HA, lane 4) compared to YY1 in the presence of MOF (lane 3; Flag IP, IB:HA). In addition, we compared the effects of overexpression and knockdown of *MOF* on YY1 degradation (Figure 2E). As expected, compared to the YY1-only group, elevated MOF facilitated YY1 degradation (IB:HA, lane 3), while knocking down *MOF* no longer promoted YY1 degradation (IB:HA, lanes 4 and 5 compared to lane 3). In a cell IF staining experiment, overexpressed MOF also led to a significant decrease in endogenous YY1 protein levels (Figure 2F). Previous experiments confirmed that the binding ability of YY1/Δ146–270 to MOF decreased significantly, suggesting an interaction between YY1 and the MOF/MSL complex. To determine whether the MOF/MSL complex regulated YY1 degradation, HCT116 cells were co-transfected with YY1, MSL2 (Figure 2G), or MSL1 (Figure 2H). MSL1 and MSL2 are both key subunits of the MSL complex. The experimental results showed that YY1 was bound to both MSL2 and MSL1 (Flag IP, IB:MSL2 or MSL1), and ubiquitin degradation of YY1 was increased by adding MSL2 or MSL1 (IB:HA). In summary, the above results indicate that the acetylation of YY1 by the MOF/MSL complex facilitated the ubiquitin–proteasome degradation pathway of YY1.

### 2.3. The Acetylation of YY1 in HCT116 Cells Was Regulated by MOF and HDAC1

It is well-known that the pan-HDAC inhibitor (pan-HDACi) SAHA (vorinostat) is involved in regulating a variety of PTMs, including ubiquitination and acetylation. For example, SAHA treatment significantly raises the global acetylation level in cells, often accompanied by changes in ubiquitination levels [34]. To further understand how the acetyl group on YY1 is removed, we first observed the effect of the HDAC inhibitor SAHA on YY1 stability. As shown in Figure 3A, overexpressed YY1 in the presence of SAHA was more prone to degradation through the ubiquitin–proteasome pathway in HCT116 cells (IB:HA, lane 4 compared to lane 2), suggesting that the higher acetylation of YY1, induced by SAHA, promoted YY1 degradation. However, the degradation of YY1/Δ146–270 was not affected by adding SAHA (Figure 3B, IB:HA, lane 5 compared to lane 4), showing the importance of the YY1/146–270 region in the acetylation-mediated proteasomal degradation of YY1. The next experimental result supports this finding. In line with the results shown in Figure 2B–E, co-transfection of MOF with YY1 promoted the ubiquitin-mediated degradation of YY1 (Flag IP/IB:HA, lane 3); however, co-transfection of YY1, MOF, and the YY1/146–270 region completely inhibited the MOF-mediated ubiquitin degradation of YY1 (Figure 3C, Flag IP/IB: HA, lane 5). The reason may be due to the competitive interaction between the YY1/146 270 region and MOF, which reduced MOF binding to YY1 (Flag IP/IB:MOF, lane 5) and led to a decrease in the acetylation-mediated ubiquitin degradation of YY1. In addition, to further specify the deacetylase that removes the acetyl group from YY1, the HDAC1 inhibitor MS275 was used in the following experiments. Overexpressed YY1, in the presence of MS275, facilitated the degradation of YY1 in a similar way to SAHA (Figure 3D, IB:HA, lane 4 compared to lane 2), indicating that HDAC1 may be the main enzyme for YY1 deacetylation. Subsequently, we tested the interaction between HDAC1 and YY1 using co-transfection and co-IP. As shown in Figure 3E, YY1 and HDAC1 mutually immunoprecipitated, indicating a binding relationship between the two proteins. We also observed the relationship between YY1, MOF, and HDAC1. As shown in Figure 3F, HCT116 cells were co-transfected with Flag-YY1/HDAC1, YY1/MOF, or YY1/MOF/HDAC1. After 48 h of transfection, proteins bound to YY1 were analyzed by Western blot. The results showed that YY1 was bound to both HDAC1 and MOF (Flag IP, lanes 3 and 4). HDAC1 and MOF may interact competitively with YY1 because, in the co-transfection of YY1/HDAC1/MOF, HDAC1 was bound to YY1 at a level remarkably lower than in the co-transfection of YY1 and HDAC1 (Flag IP, lane 5 compared to lane 3). To further verify that the acetylation of YY1 was regulated by MOF and HDAC1, we designed an additional experiment, as shown in Figure 3G. Consistent with the previous experimental results, MOF improved the acetylation of YY1 (IB:Pan-ac, lane 3), while a MOFG327E mutant could not acetylate YY1 (IB:Pan-ac, lane 5). In addition, the overexpression of HDAC1 attenuated MOF-mediated YY1 acetylation (IB:Pan-ac, lane 4 compared to lane 3). In summary, our experimental results confirmed that the dynamic changes in YY1 acetylation in cells are regulated by the MOF/MSL complex and HDAC1, and the acetylation level of YY1 determines its ubiquitin–proteasome degradation pathway.

### 2.4. The YY1K183 Site Is the Main Ubiquitylation Site That Maintains YY1 Stability

We demonstrated in the above studies that MOF binds to the 146–270 region of YY1, and without this region, acetylation-mediated degradation of YY1 was not observed, indicating that the ubiquitination site of YY1 may exist in this region. The three lysine sites—lysine 183 (K183), K208, and K258—that may be possible ubiquitin-modification sites were discovered by cautious sequence analysis and prediction using a web-based application (https://www.phosphosite.org, accessed on 5 June 2018) (Figure 4A). We first created point mutation plasmids with relevant sites in order to observe the impact of these lysine sites on ubiquitin modifications of YY1. Point mutants of YY1 protein expression were verified by a Western blot (Figure 4B). Then, HCT116 cells were transfected with an increasing amount of wild-type and mutant plasmids. As shown in Figure 4C, the ubiquitin degradation level of wild-type (lanes 2–4) and YY1K208R mutant (lanes 8–10) groups increased in a dose-dependent manner. Although overexpression of the YY1K258R group showed a similar degradation trend, it was notably weaker than those in the YY1wt group (lanes 11–13). However, compared to the YY1wt group, the YY1K183R mutant did not exhibit increased protein degradation with an increasing transfection dose (lanes 5–7). These results suggested that the YY1K183 and YY1K258 sites, and particularly K183, play an important role in the regulation of YY1 degradation. Next, we added double mutants to identify alterations in ubiquitination to further confirm the impact of the two locations: YY1K183 and YY1K258. The results obtained supported the idea that the YY1K183 and YY1K258 sites participated in YY1 ubiquitin modification (Figure 4D, lanes 11–13). It should be noted that the mutation at K183 of YY1 is sufficient to inhibit the degradation of YY1 (lanes 5–7), indicating that this site is the main ubiquitin modification site for YY1 degradation. We were interested in seeing if, following a mutation of the ubiquitination site, a change in acetylation would also result in a change in ubiquitination. Thus, we treated cells with SAHA to alter the acetylation status of YY1, and we detected ubiquitination changes in transfections of YY1wt and mutants. Consistent with previous results, the ubiquitination of YY1wt increased (Figure 4E, lane 3 compared to lane 2), while the ubiquitin-mediated degradation of the YY1K183R mutant remained unchanged (Figure 4E, lane 5 compared to lane 4). However, the YY1K258R mutant still produced an increase in ubiquitination (lane 7 compared to lane 6). Taken together, the YY1K183 site may, possibly, be the main ubiquitin degradation site regulated by acetylation. Subsequent experimental results confirmed that the protein stability of the YY1K183 mutant was improved when cells were treated with the protein synthesis inhibitor cycloheximide (CHX) in comparison to the YY1wt (Figure 4F, lanes 6–8 compared to lanes 2–4). Next, we investigated the effects of MOF-mediated acetylation on the ubiquitin-mediated degradation of the YY1K183R mutant. As shown in Figure 4G, the binding of the YY1K183R mutant to MOF was not considerably reduced, but its ubiquitin-mediated degradation was remarkably decreased compared to the YY1wt group.

### 2.5. The YY1/146–270 Region and the YY1K183 Site Interfered with p53-Mediated p21 Expression in HCT116 Cells

As a transcription factor, YY1 has both transcriptional activation and repression domains; therefore, it has the ability to activate or repress gene transcription by attracting various transcriptional cofactors to its activation or repression domains. For example, YY1 can activate cell division cycle 6 (*CDC6)*, *Snail*, *HDAC1*, and *c-MYC* gene transcription, as well as repress BCL2-associated X protein (*BAX)*, *TP53* (encoding p53), cyclin-dependent kinase inhibitor 3 (*CDKN3)*, and SRY-box transcription factor 2 (*SOX2)* genes (Figure 5A) [35]. We have confirmed, in our experiments, that the YY1/146–270 region is the MOF binding region, and the YY1K183 site may be the main site for acetylation-mediated ubiquitin degradation of YY1. To further explore the effects of mutants of the YY1/146–270 region and YY1K183R in the regulation of p53 response element (p53RE)-mediated transcription of downstream target genes, multiple p53REs contained in a pp53-TA-Luc plasmid (p53RE-Luc) (Figure 5B) were used as a model to reflect *p53* transcriptional activity. P53RE-Luc activity was determined by a dual luciferase assay. Then, the impact of YY1 on *p53* transactivation was estimated by the co-transfection of p53RE-Luc with YY1wt, YY1/146–270, or YY1/Δ146–270 plasmids. In contrast to basal level luciferase activity, YY1wt decreased p53RE luciferase activity (Figure 5C, column 3). This result supported our previous report [36]. A similar result was observed in the co-transfection of p53RE-Luc with the YY1/Δ146–270 plasmid (Figure 5C, column 5). In turn, p53RE luciferase activity was reversed by the simultaneous transfection of the YY1/146–2270 region (Figure 5C, column 4). This suggested that the YY1/146–270 region has an important role in the transcriptional regulation of YY1. Moreover, we measured the protein expression levels of p53 downstream target genes, including p21, BAX, and GADD45, using the whole cell lysates remaining after detecting luciferase activity. As shown in Figure 5D and Appendix A (quantified p21, BAX and GADD45 proteins), consistent with the transcriptional activity of p53, both YY1 full-length and YY1/Δ146–270 inhibited the downstream proteins of p53, including p21, BAX, and GADD45, while the 146–270 region did not produce the same inhibition. *YY1* knockdown experiments also supported this result. After knocking down *YY1* in HCT116 cells, the expression of p53 protein was significantly increased, and simultaneously, the expression of its downstream cyclin-dependent kinase inhibitor 1A (p21) was increased, while B-cell lymphoma 2 (Bcl2) protein levels were decreased (Figure 5E). This aroused our curiosity as to whether the YY1K183R and YY1K258R mutants also had similar regulatory effects on p53 downstream proteins. By performing Western blots, we observed that YY1wt inhibited p21 protein expression in a dose-dependent manner. The mutation, however, did not prevent the production of the p21 protein, and high dose mutants, instead, increased the expression level of the p21 protein (quantified protein levels of p21 are marked below the p21 signals, Figure 5F, IB:p21). These results implied that amino acid substitutions lessened YY1 suppression of *TP53* transcriptional activity. We are aware that MOF controls YY1 acetylation-mediated ubiquitin degradation; therefore, we wondered whether MOF affected the repression of the transcriptional activity of *p53* by YY1. Our analysis of p53RE-mediated luciferase activity revealed that luciferase activity decreased with YY1 in a dose-dependent manner. Unexpectedly, overexpressed MOF dramatically enhanced p53RE-mediated luciferase activity. However, this enhancement was suppressed by co-transfection with MOF and YY1 (Figure 5G). Taken together, we hypothesized that YY1 and MOF may jointly participate in regulating the transcriptional activation of downstream target genes mediated by p53.

### 2.6. A YY1K183R Mutant Inhibited CDC6 Transactivation, and Suppressed the Proliferation of HCT116 Cells

*CDC6* is another familiar downstream target gene of YY1, which binds to the upstream of the *CDC6* transcriptional start site to initiate *CDC6* gene transcription at the G1-S transition [37]. The binding site of YY1 on the *CDC6* promoter region is shown in Figure 6A. To determine the impact of YY1 and its mutants on the CDC6 expression level, HCT116 cells were transiently transfected with YY1wt and YY1 mutants. The next day, the cells were treated with 1 mM hydroxyurea for 16 h. The expression levels of the CDC6 protein are shown in Figure 6B and Appendix A (quantified CDC6 protein). It was found that YY1wt dose-dependently promoted CDC6 protein expression levels (lanes 3 and 4). Conversely, YY1 mutants, including YY1K183R, YY1K258R, YY1K183 + 258R, and YY1/Δ146–270, did not promote an increase in the expression level of the CDC6 protein in a dose-dependent manner (lanes 5–12). To further confirm these results, HCT116 cells were co-transfected with pGL3-CDC6-Luc, as well as YY1wt or YY1K183R mutant plasmids, and the whole cell lysates were collected at 48 h to estimate their luciferase activities. As shown in Figure 6C, the luciferase activity of CDC6-Luc increased, in a dose-dependent manner, only when the pGL3-CDC6-Luc and YY1wt plasmids were co-transfected, indicating the role of the YY1K183 site in regulating transactivation of *CDC6* (columns 3–4). Furthermore, CDC6 proteins were detected after the co-transfection of pGL3-CDC6-Luc with YY1wt, YY1K183R, or YY1K258R plasmids (Figure 6D). A dose-dependent increase in CDC6 protein levels was only seen in YY1wt transiently transfected HCT116 cells (lanes 2 and 3) but not in cells with overexpression of YY1K183R and YY1K258R plasmids (lanes 4–7). As an oncogene, CDC6 is highly expressed in tumor tissues, driving tumor cell DNA replication and unlimited proliferation [38]. As shown in Figure 6E, we analyzed the effects of YY1wt and the YY1K183R mutant on HCT116 cell proliferation with an EdU proliferation assay. The statistics of the cell proliferation rate are shown in Figure 6F. The results demonstrated that the transient transfection of YY1wt, but not the YY1K183R mutant, promoted cell proliferation (** *p* < 0.01). Corresponding YY1wt and YY1K183R protein expression levels are shown in Figure 6G. By using MTT and colony formation experiments, we also observed the differences in the effects of YY1wt, as well as YY1K183R or YY1K258R mutants, on cell survival and the clonogenic ability in HCT116 or SW480 colon cancer cells. In MTT assays, YY1 increased the viability of HCT116 cells, showing a significant difference compared to the control group on day 7, while the mutant inhibited YY1 from promoting cell viability (Figure 6H). Colony formation experiments were conducted in HCT116 and SW480 cells. As shown in Figure 6I, YY1 greatly enhanced the number of clones formed in both cell types, while the YY1K183 mutant significantly antagonized the effect of YY1. In addition, both the YY1wt and YY1K183 mutant, when combined with MOF, significantly inhibited clone formation compared to the YY1wt or YY1K183R mutant groups alone. Figure 6J shows the quantified clone numbers in HCT116 and SW480 cells.

## 3. Discussion

YY1 is a ubiquitously expressed transcription factor that is frequently highly expressed in a variety of tumors, indicating its importance in the occurrence and development of tumors. In line with this, overexpression of YY1 dramatically increased cell viability in HCT116 cells, as well as the clonogenic ability of HCT116 and SW480 cells, in our experiments. A large amount of research data shows that YY1 is not only a potential target for different PTMs but also that the type of PTM is closely related to its transcriptional activity [39]. Given that the YY1 protein sequence contains disordered regions, its complete structure has not been resolved, which also implies the functional complexity and diversity of YY1 [40]. Our study revealed a new protein interacting with YY1, the acetyltransferase MOF, and we confirmed that MOF acetylates YY1 and, at the same time, promotes the ubiquitin-mediated degradation of YY1. In addition, we identified a novel K183 site that mediated YY1 ubiquitination. Point mutations at this site suppressed the transcriptional activity of YY1. Moreover, the acetylation of YY1 by MOF promoted the p53-mediated transactivation of downstream target genes.

Both the acetylation and ubiquitination of proteins occur on lysine residues, so a previous lysine acetylation can influence a subsequent protein ubiquitination. Therefore, it can be inferred that the stability of proteins is regulated by both the ubiquitin–proteasome pathway degradation and the acetylation-based promotion or protection of lysine ubiquitin modifications [30]. In our research, we observed that MOF acetylated YY1, but it did not prevent ubiquitination (Figure 2), although in most cases, acetylation of the protein prevented its degradation by ubiquitin. In actuality, the acetylation-dependent regulation of protein stability involves a complex process. In some cases, protein acetylation also accelerates protein ubiquitin degradation. For example, acetylated hypoxia-inducible factor-1α (HIF1α) increases its interaction with E3 ubiquitin ligase von Hippel-Lindau (VHL); the acetylated retinoblastoma (RB) tumor suppressor protein also recruits murine double minute 2 (MDM2), which, in turn, degrades the substrate p53 by ubiquitination. Additionally, acetylated heat shock protein 90 (HSP90) weakens its interaction with molecular chaperones that are modified by ubiquitination [30]. Recently, research has shown that general control non-depressible 5 (GCN5) HAT acetylated p62, and the acetylated p62 increased interactions with the E3 ligase, Kelch-like ECH-associated protein 1 (Keap1), to promote proteasome-mediated ubiquitin degradation [41]. Regarding our experimental results, we put forward two conjectures: (1) we found that overexpression of MOF promoted the expression of endogenous Smurf 2 protein (Figure 2A), suggesting that MOF regulated the known E3 ligase Smurf 2 of YY1, thereby promoting the ubiquitination of YY1; (2) we determined that both MSL1 and MSL2 interacted with YY1 and promoted its ubiquitination (Figure 2F,G), suggesting that the degradation of YY1 is related to the MOF/MSL complex. MSL2 with a RING domain is an E3 ubiquitin ligase, and it can target p53 to promote MDM2-independent cytoplasmic localization [42]. Therefore, we speculated that the MSL complex was bound to the 146–270 region of YY1 (Figure 1E) and participated in regulating YY1 stability by modulating the acetylation–ubiquitin axis. However, more research is required to determine the precise mechanism of MOF-mediated YY1 ubiquitination.

Previous studies have shown that YY1 repressed p53-mediated transcriptional regulation [36]. There is physical interaction of YY1 with MDM2 and p53, and YY1 was shown to promote MDM2-mediated ubiquitin degradation of p53 [43]. In our experiments, we found that YY1 inhibited p53-mediated transactivation, leading to a decrease in the expression of the downstream target gene *CDKN1A* (encoding p21). However, transient transfection of the YY1/146–270 region reversed the inhibition of YY1 on p53-mediated transactivation, resulting in an increase in p21 protein expression (Figure 5E,F). According to the published literature, YY1/146–270 contains a binding region for MDM2. Thus, overexpression of this region may result in a large number of MDM2 molecules binding to YY1/146–270, leading to a decrease in MDM2 binding to the full length YY1, weakening the inhibitory effect of YY1 on MDM2-mediated p53 ubiquitin degradation, thus increasing the transcriptional expression of p21. Additionally, overexpression of the YY1K183R mutant also facilitated the p21 protein level in HCT116 cells (Figure 5G). It is possible that the point mutation may alter the protein conformation of YY1 and weaken the interaction of YY1 with p53, thereby attenuating the inhibition of p53 transactivation by YY1 and leading to an increase in the expression of the downstream gene *CDKN1A* (p21). Interestingly, following MOF overexpression, we found a dose-dependent p53-mediated increase in luciferase activity (Figure 5G). YY1 (200–295 aa) were not only bound to p53 [43] but also interacted with MOF in its 146–270 region. Therefore, we speculated that p53 and MOF competed to bind to YY1. Overexpression of MOF weakened the interaction between YY1 and *TP53*, releasing the transcriptional inhibition of YY1 on *TP53*, resulting in a significant increase in p53-Luc luciferase activity. However, we need to determine whether MOF acts as a transcription cofactor in the MSL complex and whether it is co-recruited with YY1 to the promoter region of *TP53*, which requires additional experimental evidence.

*CDC6* is another familiar downstream target gene of YY1, which binds to the promotor region of *CDC6* to initiate *CDC6* transcription at the G1-S transition [35]. Overexpression of YY1 revealed a dose-dependent increase in CDC6-Luc luciferase activity; however, the YY1K183R mutant no longer activated the transcription of *CDC6* (Figure 6C). Further research has confirmed that the mutation at the K183 site not only altered the original transcriptional activity of YY1 but also affected its biological functions. As expected, overexpression of YY1 remarkably increased the clonogenic ability of both HCT116 and SW480 colon cancer cells; however, the YY1K183R mutant lost its ability to promote colony formation (Figure 6I). Based on these studies, we speculate that the K183 site may regulate the biological functions of YY1 by modulating its protein stability, which is controlled by the acetylation–ubiquitin proteasome degradation pathway. Although we did not conduct in vitro ubiquitin assay or mass spectrometry analysis, the YY1 degradation did change due to its acetylation status in the presence of proteasome inhibitor MG-132. At the same time, the YY1K183 mutant inhibited the degradation of YY1, stimulated by acetylation induced by SAHA or MOF (Figure 4), suggesting that MOF-mediated YY1 acetylation may stimulate its degradation, mainly, through K183 site. Of course, the other degradation pathways of YY1 cannot be ruled out.

## 4. Materials and Methods

### 4.1. Antibodies

Anti-Flag (M2) and anti-Myc-agarose, as well as anti-Flag M2 (F3165) mouse monoclonal antibody and hydroxyurea (HU, H8267) obtained from Sigma-Aldrich (St. Louis, MO, USA). Anti-YY1 (H414; sc-1703) rabbit polyclonal, anti-CDC6 (sc-13136), anti-Smurf2 (sc-393848), and anti-Myc (9E10) mouse monoclonal antibodies were from Santa Cruz Biotechnology (Dallas, TX, USA). Anti-MSL1 (24373-1-AP) mouse monoclonal, anti-p21 (10355-1-AP), and anti-BCCIP (16043-1-AP) rabbit polyclonal antibodies were purchased from Proteintech (Wuhan, China). Anti-MOF (A3390) polyclonal antibody was from ABclonal Technology (Wuhan, China). Pan-ac (PTM0105RM) polyclonal antibody was from Jingjie Biotechnology (Hangzhou, China). Anti-MSL2 (ab83911) was from Abcam (Shanghai, China). Anti-GADD45 (RLT1832), Bcl2 (RLM3041), and anti-HA (RLM3003) mouse monoclonal antibodies were obtained from Ruiying Biological (Suzhou, China). Anti-p53 mouse monoclonal antibody was provided by Boster Group (BM0101, Wuhan, China). Anti-GAPDH (NM_002046, full length) rabbit polyclonal antibodies were raised against bacterially expressed proteins (Jilin University, Changchun, China). MG-132 (I-130) was purchased from Boston Biochem (Cambridge, MA, USA). Cycloheximide (CHX, DH466-1) was from Beijing Dingguo Changsheng Biotechnology Co., Ltd. (Beijing, China). HAT inhibitor MG149 (S7476) and HDAC inhibitor vorinostat (SAHA, S1047) were from Selleck Chemicals (China, Shanghai).

### 4.2. Cell Culture

HCT116 and SW480 cell lines were purchased from the Shanghai Biotechnology Co., Ltd. (Shanghai, China) and Procell Life Science & Technology Co., Ltd. (Wuhan, China). Each cell line was authenticated via short tandem repeat (STR) profiling within 3 years. All experiments were performed with mycoplasma-free cells. HCT116 colon cancer cells were cultured in RPMI-1640 medium (Meilunbio^®^, Dalian, China), and SW480 colon cancer cells and human embryonic kidney 293T cells were cultured and maintained in Dulbecco’s modified Eagle’s Medium (DMEM; Meilunbio^®^, Dalian, China), containing 10% fetal bovine serum (FBS, Procell, Wuhan, China) and 1% penicillin-streptomycin mixture (P/S; Solarbio Science & Technology, Beijing, China) at 37 °C in the presence of 5% CO_2_.

### 4.3. Plasmids and Transient Transfection

Full-length cDNAs encoding the human MSL1 (NM_001012241), MOF (NM_032188), MSL2 (NM_018133.4), HDAC1 (NM_004964.3), and YY1 (NM_003403) proteins, different truncations including YY1 (1–146 and YY1Δ146–270 aa), different point mutants including YY1 (K183R, K208R, K258R, K183/258R), and MOFG327E were subcloned into a pcDNA3.1 (–) vector with Flag or Myc tags. The plasmids were transiently transfected into cells using polyethyleneimine (PEI) (23966, PolySciences, Beijing, China), according to the manufacturer’s instructions.

### 4.4. Immunoprecipitation (IP)

HCT116 or HEK 293T cells, cultured in 10 cm tissue culture plates, were transiently transfected with the above Flag or Myc-tagged plasmids. Then, 48 h after transfection, cells were collected with RIPA lysate buffer containing 1% NP-40, 150 mM NaCl, 50 mM Tris-HCl, 10% glycerol, 1 mM dithiothreitol (DTT), and complete protease inhibitor cocktails. Prepared whole-cell lysates were incubated overnight at 4 °C with anti-Flag (M2) or anti-Myc-agarose beads. The immunoprecipitated proteins were eluted with a 4 × SDS buffer, and bound proteins were analyzed by a Western blot with anti-Flag or anti-Myc antibodies.

### 4.5. Expression of Recombinant Proteins in Escherichia coli

Full-length or deletion mutants of YY1 (1–146, 1–213, 1–270, 146–270, 146–414, 207–414, 270–414 aa residues, and YY1Δ146–270) were subcloned into pET41a vector. His-GST tagged different YY1 truncated proteins that were expressed from pET41a in BL21 (DE3) Codon Plus *Escherichia coli*. bacterial cells.

### 4.6. MTT Assay

HCT116 cells (2000 cells/well) were seeded in 96-well plates, and the cell viability was measured by incubating cells with the CellTiter 96^®^Aqueous One Solution Cell Proliferation Assay Kit (G3580, Promega Corporation, Madison, WI, USA) at 24 h, 72 h, day 5, and day 7, respectively. The absorbance at a wavelength of 450 nm was measured using a microplate reader (Infinite F200 Pro, TECAN, Shanghai, China).

### 4.7. Colony Formation Assay

HCT116 and SW480 cells, grown to ~30% confluence in 6-well plates, were transfected with wild-type or point mutants of YY1. Cells were digested with trypsin 48 h later, they were resuspended with RPMI-1640 medium, and split into a new 6-well plate with 2000 cells/well. A week later, the cells were fixed with 4% paraformaldehyde, at room temperature, for 15 min, and formed colonies were stained with 0.1% crystal violet for 20 min. Colonies containing >20 cells were scored as positive. Significant differences between pcDNA3.1 control and Flag-YY1 or Flag-YY1 mutant groups were analyzed via the Student’s *t*-test. The statistical difference of *p* < 0.05 was considered to indicate a statistically significant result.

### 4.8. Immunofluorescence Staining

HCT116 cells, grown to ~30% confluence in 24-well plates containing a coverslip (8D1007, Nest) on each well, were transfected with pcDNA3.1 and Flag-MOF plasmids. Additionally, 48 h after transfection, cells were immunostained with YY1 (1:400 dilution) and MOF (1:200) primary antibodies, and FITC-conjugated secondary antibodies (1:300, Santa Cruz sc-2012) were used. Cell nuclei were stained with Vectashield with DAPI (H-1200) (Vector Laboratories, Inc., Burlingame, CA, USA). Fluorescent images were observed with an Olympus BX40F Microscope (Olympus Corporation, Miyazaki, Japan) with a silicon immersion 40× objective.

### 4.9. EdU Assay

EdU incorporation assay was performed using a BeyoClick™ EdU Cell Proliferation Kit with Alexa Fluor 488 in vitro Imaging Kits (Beyotime C0071s, Shanghai, China). Briefly, HCT116 cells were transiently transfected with pcDNA3.1, F-YY1, and F-K183R. The cells were re-plated on a 12-well plate 48 h later. After 24 h, 10 μM of EdU (5-ethynyl-2′-deoxyuridine) was added and incubated in a 37 °C incubator for 2 h. The cells were then fixed with 4% paraformaldehyde for 15 min and rinsed with PBS containing 0.5% Triton-X-100. The Hoechst33342 (GC10939, GLPBIO) was used to stain the nuclei. The cell proliferation rate was calculated according to the manufacturer’s recommendations (BeyoClick EdU-488 Kit, C0071s, Shanghai, China). Using a fluorescent microscope, pictures of three randomly selected regions of each group were captured.

### 4.10. Luciferase Reporter Assay

Pp53-TA-Luc plasmid (D2223, Beyotime), including multiple p53 response elements (ACGTTTGCCTTGCCTGGACTTGCCTGGCCTTGCCTTGGACATGCCCGGGCTGTC), was obtained from Beyotime Biotechnology (Shanghai, China). However, the promoter region of CDC6 (−1066~+240bp) was introduced into pGL3-Luc vector (Promega Corporation). HCT116 cells grown on 12-well plates were co-transfected with 0.4 μg pGL3-CDC6-Luc and pGL3-P53RE-Luc, which encodes firefly luciferase, 0.12 ng of the control plasmid Renilla luciferase vector, and the plasmids expressing Flag-YY1, YY1K183R, or Flag-MOF using PEI reagent. Total effector plasmids in each transfection were adjusted to 0.8 μg with empty vectors. Around 48 h after transfection, the luciferase activities of p53-Luc and pGL3-CDC6-Luc were measured using the Dual-Luciferase Reporter assay kit (Promega, Madison, WI, USA), and the renilla luciferase activity was used as the control for normalization.

### 4.11. Statistical Analysis

The data were analyzed using SPSS 16.0 software (IBM, Armonk, New York, NY, USA). Results are expressed as the means and standard deviation (SD). Student’s *t*-test was used to analyze the difference between two independent samples. The one-way analysis of variance (ANOVA) with post-hoc test was used to measure differences among the multiple groups. A statistically significant difference was considered to be present at *p* < 0.05.

## Figures and Tables

**Figure 1 ijms-24-08719-f001:**
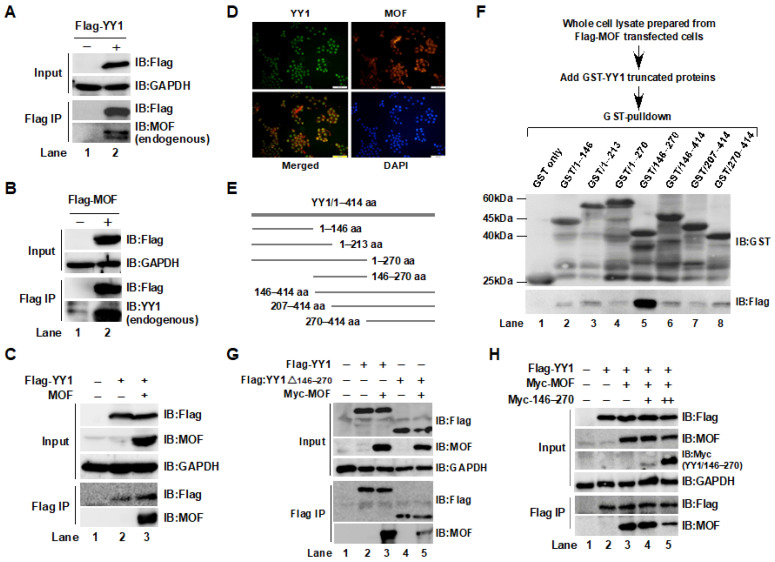
Immunoprecipitation and GST-pulldown experiments confirmed the binding of MOF and YY1. (**A**,**B**) Endogenous MOF and YY1 were immunoprecipitated by overexpressed Flag-YY1 and Flag-MOF in HCT116 cells. Bound proteins were measured by Western blot analysis. GAPDH was used as an internal control. (**C**) Co-transfection and Flag IP confirmed the interaction between YY1 and MOF. HCT116 cells were co-transfected with Flag-YY1 and untagged MOF, and bound MOF was measured after a Flag IP. (**D**) The colocalization analysis of YY1 and MOF. Endogenous YY1 (green) and MOF (red) in SW480 cells were visualized by IF staining. DAPI staining showed the nuclei. Scale bar indicates 200 µm. (**E**) Different lengths of YY1. (**F**) A GST-pulldown assay was performed by mixing whole cell lysates of Flag-MOF overexpressing 293T cells and the GST-tagged deletion mutants of YY1 proteins. (**G**) The lack of the 146–270 region in YY1 decreased its binding to MOF. (**H**) The 146–270 region of YY1 competed for binding to MOF. HCT116 cells were transiently transfected with YY1 and MOF, as indicated; 48 h later, the MOF protein level was analyzed after a Flag IP. IB, immunoblot.

**Figure 2 ijms-24-08719-f002:**
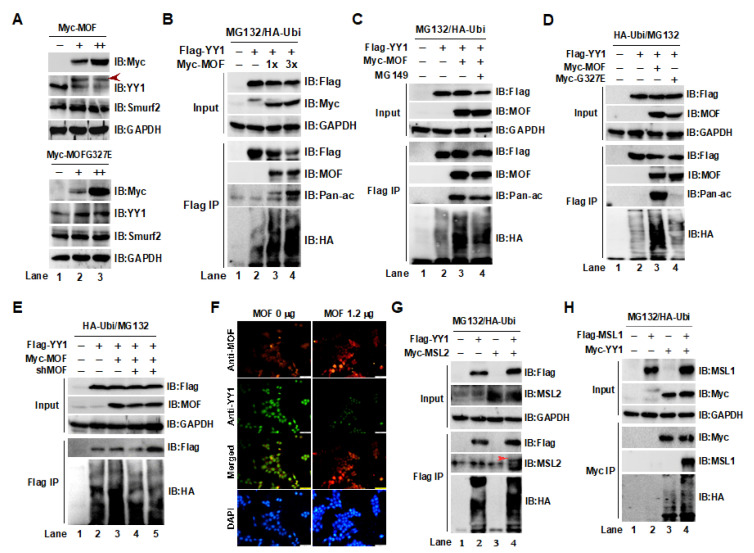
Acetylation of YY1 by the MOF/MSL complex negatively regulated its protein stability in HCT116 cells. (**A**) YY1 may be modified by MOF. Endogenous YY1 was detected by a Western blot analysis after overexpressing MOFwt or the MOFG327E mutant in HCT116 cells. The red arrow indicates YY1 that may be modified. (**B**) Acetylation of YY1 by MOF facilitated its degradation. (**C**) Degradation of YY1, caused by MOF, was inhibited by the MOF enzyme activity inhibitor MG149. (**D**) The inactivated MOFG327E mutant inhibited the degradation of YY1. (**E**) The knockdown of *MOF* reduced YY1 degradation. (**F**) Elevated MOF decreased the endogenous YY1 protein level. Scale bar indicates 200 µm. (**G**,**H**) The MOF/MSL complex promoted YY1 degradation. HCT116 cells were transfected with YY1 alone or co-transfected with MSL2 or MSL1, and 48 h later, YY1 degradation was measured by Western blot using an anti-HA antibody. The red arrow in (**G**) indicates MSL2 protein. GAPDH was used as an internal control.

**Figure 3 ijms-24-08719-f003:**
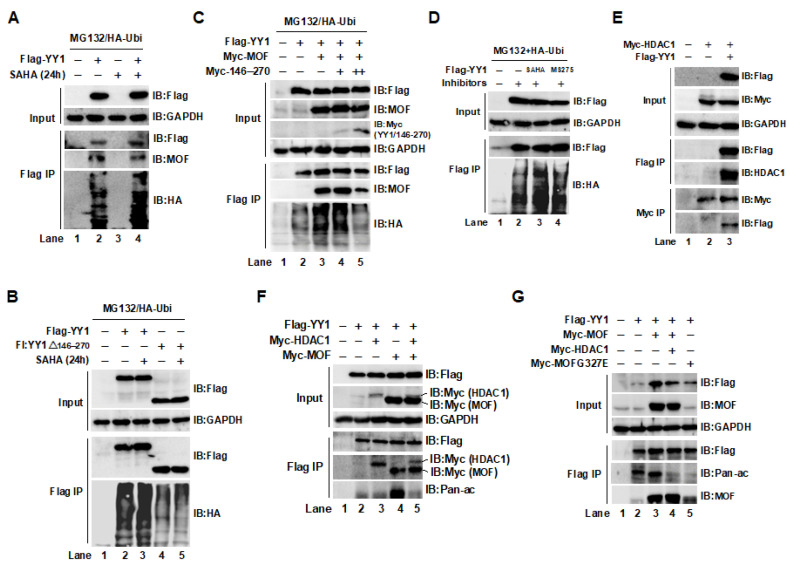
MOF-mediated acetylation of YY1 is inhibited by HDAC1 in HCT116 cells. (**A**) The HDAC inhibitor, SAHA, increased YY1 degradation. (**B**) The degradation of YY1 without the 146–270 region was not affected by SAHA. (**C**) The interaction of MOF with the YY1/146–270 region controlled YY1 stability. (**D**) Both the SAHA and the HDAC1 inhibitor, MS275, promoted YY1 degradation. (**E**) YY1 and HDAC1 were bound to each other. (**F**) MOF and HDAC1 may competitively bind to YY1. (**G**) YY1 acetylation in cells was regulated by MOF and HDAC1.

**Figure 4 ijms-24-08719-f004:**
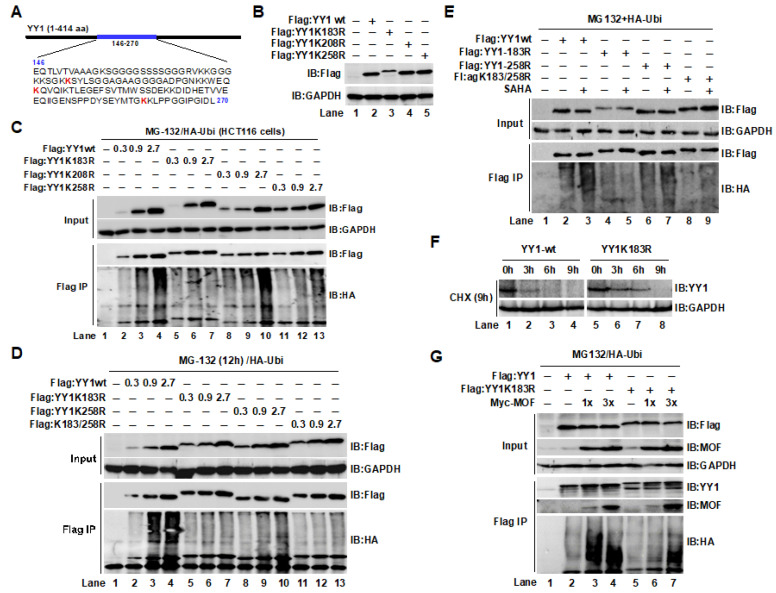
The YY1K183 site regulated the acetylation-mediated ubiquitin degradation of YY1. (**A**) There are three predicted potential ubiquitination sites (K183, K208, and K258 indicated by red color) in the 146–270 region of YY1. (**B**) Protein expression of the mutant plasmids. (**C**) The effect of mutations of lysine K183, K208, and K258 to arginine on YY1 ubiquitination. (**D**) Lysine K183 and K258 mutations suppressed YY1 ubiquitin degradation. (**E**) The effect of acetylation on ubiquitination of YY1 and mutants. (**F**) Protein stability of the YY1K183R mutant. (**G**) The effects of MOF-mediated acetylation on the ubiquitin-mediated degradation of the YY1K183R mutant.

**Figure 5 ijms-24-08719-f005:**
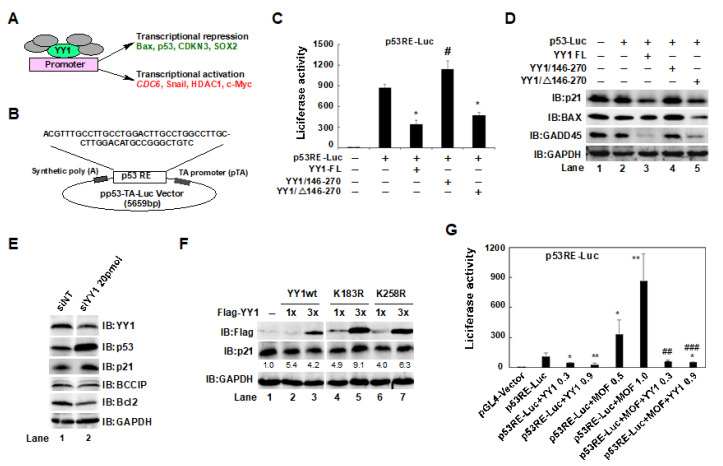
The YY1/146–270 region and the YY1K183 site play important roles in p53RE-mediated downstream target gene transactivation in HCT116 cells. (**A**) The schematic diagram of target genes up- or down-regulated by YY1. (**B**) The schematic diagram of a p53RE-Luc plasmid. Multiple response elements of *TP53* were inserted into the pp53-TA-Luc vector. (**C**,**D**) The effects of YY1 and its truncated mutants on p53RE-Luc luciferase activity and related protein levels. * *p* < 0.05 or # *p* < 0.05, compared to the p53RT-Luc group. (**E**) P53 and its downstream protein levels in *YY1* knockdown HCT116 cells. (**F**) Effects of YY1 and its point mutants on p21 protein levels. Increasing amounts of YY1wt, YY1K183R, and YY1K258R were transfected into HCT116 cells, and p21 protein levels were analyzed by a Western blot using the anti-p21 antibody. (**G**) The effects of YY1 and MOF on p53RE-Luc luciferase activity. * *p* < 0.05 or ** *p* < 0.01 compared to the p53RT-Luc group; ## *p* < 0.01 or ### *p* < 0.001 compared to the p53RT-Luc+MOF groups.

**Figure 6 ijms-24-08719-f006:**
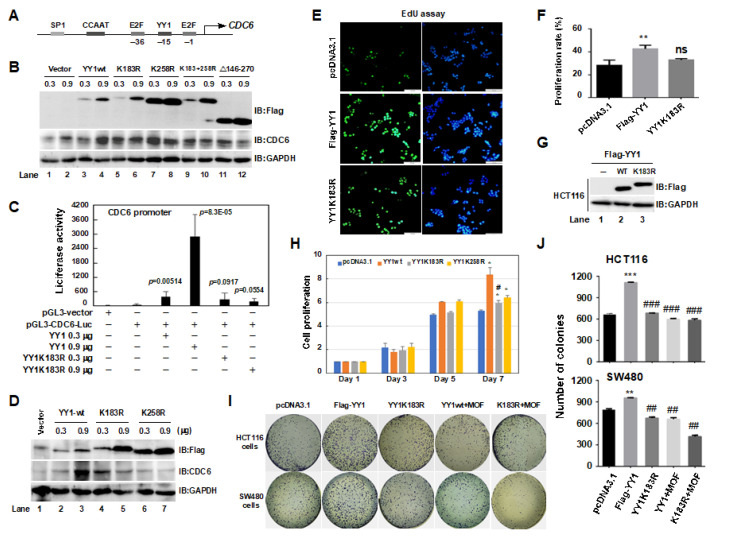
YY1K183R/K258R mutants suppressed CDC6 transactivation and inhibited cell proliferation in HCT116 and SW480 cells. (**A**) A schematic diagram of YY1 in the promoter region of *CDC6*. (**B**) The effects of YY1wt and its mutants on CDC6 protein levels. (**C**) The effects of YY1wt and its mutants on CDC6-Luc luciferase activity. (**D**) The effects of YY1wt and its mutants on CDC6 protein expression levels. (**E**–**G**) The effects of YY1wt and YY1K183R on cell proliferation, as detected by an EdU proliferation assay. The proliferation rate is shown in (**F**) (** *p* < 0.01 compared to the pcDNA3.1 group; ns, no significant difference between the pcDNA3.1 and YY1K183R groups), and the YY1 protein levels are revealed in (**G**). (**H**) MTT assays. * *p* < 0.05 compared to the pcDNA3.1 group; # *p* < 0.05 compared to the YY1wt group. (**I**) The effects of YY1 and mutants on cell clone formation. (**J**) Quantified colony numbers for the experimental results in. ** *p* < 0.01, *** *p* < 0.001 compared to the pcDNA3.1 group; ## *p* < 0.01 or ### *p* < 0.001 compared to the Flag-YY1 group.

## Data Availability

No new data were created.

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
