# Peer review of "The Males Absent on the First (MOF) Mediated Acetylation Alters the Protein Stability and Transcriptional Activity of YY1 in HCT116 Cells"

_ijms, 2023, doi:10.3390/ijms24108719_

Round 1

Reviewer 1 Report

The authors demonstrated that histone acetylase MOF can acetylate the YY1 transcription factor and that this acetylation stimulates the degradation of this factor. Degradation is suppressed by an inhibitor of proteasome activity, from which the authors conclude that it is regulated by ubiquitination, which, in turn, is regulated by acetylation. Using deletion analysis, the authors identified a YY1 polypeptide chain segment essential for interaction with MOF. The results presented in the article are of potential interest. However, many conclusions of the authors are not directly confirmed by experimental data. For this reason, the article should be significantly improved before considering for publocation.

My main concern is that the authors nowhere directly analyze YY1 ubiquitination, but judge it only on the basis of the level of protein degradation. In this regard, one can speak of ubiquitinylation only in the suggestive mood. Thus, the replacement of lysine with another amino acid at position 183 suppresses the degradation of YY1 stimulated by acetylation. The authors conclude that this site is ubiquitinated, whereby the protein is directed to the proteasome. However, it can equally be assumed that acetylation at this site stimulates ubiquitination in some other domain of the protein. Until ubiquitination at certain sites, including the K183 site, is not directly shown (eg, by mass spectrometric analysis), all conclusions concerning the particular ubiquitination sites are nothing more than assumptions. And this should be written directly in the manuscript.

Other notes:

Figure 1D: It is only possible to say that both MOF and YY1 are present in the nuclei. Much higher magnification/resolution should be used to analyze colocalization. Besides, the size of scale bar should be indicated

Fig 1G: All designations should be explained in the legend. What is the meaning of designation “IB”. If “IB:MOF” is MOF band visualized by immunostaining, than I do not see significant reduction of MOF binding to YY1d146-270 compared to wilt type YY1. Experiment should be repeated several times and results should be quantified.

Fig 4B: Why YY1K183R has a reduced mobility in PAAG compared to other mutants and wild type YY1? May it happen that additional changes occurred during the preparation of this mutant? This variant is important for further analysis and the amino-acid sequence should be verified.

Reviewer 2 Report

Tha article "MOF-mediated acetylation alters the protein stability and transcriptional activity of YY1 in 4 HCT116 cells"  describes the region of YY1 necessary for MOF-mediated regulation and its effects on cell proliferation. The article is interesting and well written. However I have some concerns:

1.- cancer cells is repeated in line 101 (at the endo of introduction section)

2.- Quantification and statistical analysis of western blot should be added

3.- Cells used should be authentified to be sure that authors works with human and HCT116 and SW480 cells

4.- ANOVA and post hoc statistical analysis is the appropiate statistical test when authors have more than two experimental groups

Round 2

Reviewer 1 Report

I cannot agree with the supposition that the 3D structure of the mutated YB1 can affect its mobility is SDS PAAG. Indeed, before the electrophoresis the samples are heated with SDS and b-mercaptoethanol to destroy the 3D structure of proteins. 

Reviewer 2 Report

Many of omy concerns have been answered. However, authors should clarify when HCT116 and SW680 cells were authentified and the number of passages and how long have they been in cultive since the authentification. Please add the information about authentification in methods section.
